# Angle-based wavefront sensing enabled by the near fields of flat optics

Soongyu Yi[1], Jin Xiang[1], Ming Zhou [1], Zhicheng Wu [1], Lan Yang[2] & Zongfu Yu [1]✉

There is a long history of using angle sensors to measure wavefront. The best example is the Shack-Hartmann sensor. Compared to other methods of wavefront sensing, angle-based approach is more broadly used in industrial applications and scientific research. Its wide adoption is attributed to its fully integrated setup, robustness, and fast speed. However, there is a long-standing issue in its low spatial resolution, which is limited by the size of the angle sensor. Here we report a angle-based wavefront sensor to overcome this challenge. It uses ultra-compact angle sensor built from flat optics. It is directly integrated on focal plane array. This wavefront sensor inherits all the benefits of the angle-based method. Moreover, it improves the spatial sampling density by over two orders of magnitude. The drastically improved resolution allows angle-based sensors to be used for quantitative phase imaging, enabling capabilities such as video-frame recording of high-resolution surface topography.

[1] Department of Electrical and Computer Engineering, University of Wisconsin-Madison, Madison, WI, USA. [2] Department of Electrical and Systems Engineering, Washington University in St. Louis, St. Louis, MO, USA. ✉email: zyu54@wisc.edu

Measuring the wavefront of light has broad application in optical characterization. Wavefront cannot be directly measured, because it is determined by phase. There are two types of indirect methods to measure the wavefront. The first is based on interference as shown in Fig. 1a. A secondary reference wave, often a replica of the primary incident wave, is brought in to interfere with the primary wave. The resulting interference patterns are used to measure the phase[1] or the gradient of the phase[2]. It offers high spatial resolution for quantitative phase imaging[1–3]. However, the interference method has certain limitations related to delicate interference setups and complex signal acquisition schemes, making it sensitive to temporal perturbation or spatial alignment. Recent work has shown exciting progresses in overcoming these shortcomings[4]. Despite rapid progress in the interference method, a second method based on angle measurement is much more dominant in commercial applications because of its robustness. Figure 1b illustrates the working principle of angle-based wavefront sensors. Arrays of micro-lenses sample the incident angles $\theta(x, y)$ on a set of grid points. These angles together can determine the wavefront[5–7]. Shack–Hartmann wavefront sensors[8,9], one type of angle sensor, have been widely used in adaptive optics such as astronomy[10,11] and biomedical imaging[12–14]. It is also widely used in metrology[15–18]. While angle-based method is fast and robust, they have a long-standing difficulty. Its spatial resolution is low, orders of magnitude lower than that of interference methods. As such, they are mostly used for slowly varying wavefronts and are not suitable for demanding applications such as quantitative phase imaging. In addition to these two main methods, there are other innovative approaches such as spatial wavefront sampling[19].

In this work, we demonstrate a high-resolution wavefront sensor based on integrated angle sensors. It has a high spatial resolution and high dynamic range. It can measure high-resolution surface morphology better than most state-of-the-art commercial tools including white light interferometry (WLI). Its fast speed and robustness also make it possible to have real-time record of the temporal dynamics of surface morphology, which used to be extremely difficult to achieve with traditional metrology tools.

## Results

**Working principle of angle-based wavefront sensing.** The low spatial resolution of angle-based wavefront sensors is caused by the large size of the lenses. One cannot decrease the size of lens indefinitely, because increased diffraction in small lenses reduces their capability to focus. Recently, we developed an ultra-small angle sensor by exploiting the near-field coupling effect between nanoscale resonators[20]. Although it, in principle, can enable high-resolution wavefront sensing, the approach cannot leverage today's advanced image sensor technology and thus it is difficult to scale to large arrays. Here we use an angle sensor based on flat optics. Flat optics[21,22] has been used for lenses, holography, and on-chip optical processing[23–26]. Recently, the integration of flat optics directly with sensor arrays enable new functionalities, such as ultra-compact spectrometers[27], interference-based wavefront sensors[2], and metasurface sensor[28,29]. Unlike most flat optics that relies on far-field effect, we explore the near-field effect of flat optics. We use the energy distribution of optical field right after a flat optics component to measure the incident angle.

Figure 2 shows the working principle of the angle sensor. We first illustrate the idea in two-dimensional space. We consider an aperture illuminated by a plane wave (Fig. 2a). Light passing through the aperture substantially deviates from a plane wave, in particular when the aperture size is comparable to the wavelength. Two edges of the aperture produce scattering waves with a phase difference determined by the incident angle $\theta$. Consequently, the scattered waves lead to a near-field distribution that is a strong function of the incident angle. To quantify this dependence, we draw a line along the center of the aperture and compare the optical energy on both sides. Rayleigh–Sommerfeld diffraction theory is used to calculate the field after the aperture (see Supplementary Section 1). At the normal direction, there is an equal amount of energy on both sides. When the incident angle tilts toward one side, energy is more concentrated on the other side. A monotonic relationship between the ratio of energy and the incident angle holds up to 45° (Fig. 2b). The local energy distribution can be measured by placing two photodetectors immediately after the aperture. As the photocurrent is determined by $\int \sigma E^2 dV$, where $\sigma$ is the effective conductivity of the material, the ratio of photocurrents directly measures the energy ratio. The monotonic relationship also holds well for apertures with the finite thickness. It has weak wavelength dependence, allowing it for broadband operations (see Supplementary Section 2).

We can understand the field by using the total-field scattered-field method (Fig. 2c, d). If without the aperture, a perfect metal mask will simply reflect the incident wave. We will have interference built up in front of the perfect metal and

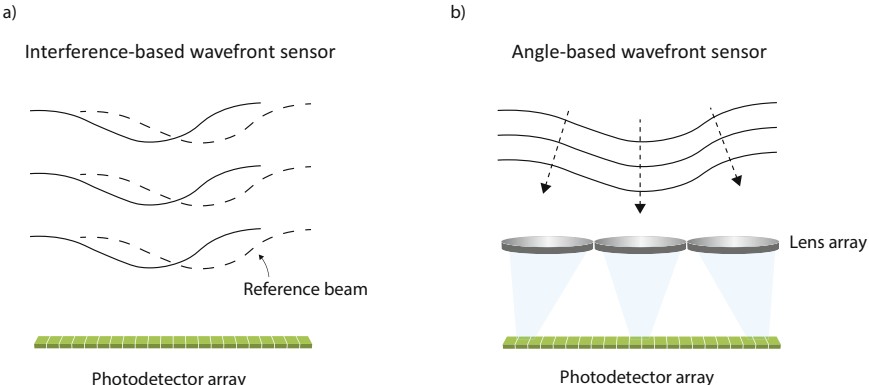

a) Interference-based wavefront sensor

Reference beam

Photodetector array

b) Angle-based wavefront sensor

Lens array

Photodetector array

**Fig. 1 Two different types of wavefront sensing. a** Interference between the incident wave and a reference wave produces intensity modulation that can be recorded by photodetector arrays. The reference wave can often be generated from the same incident wave. Multiple interference patterns are usually needed in order to retrieve the wavefront without ambiguity. **b** Arrays of micro-lenses are used to measure the spatial distribution of incident angles on the wavefront. The size of lenses is usually much larger than the optical wavelength in order to have a good focus.

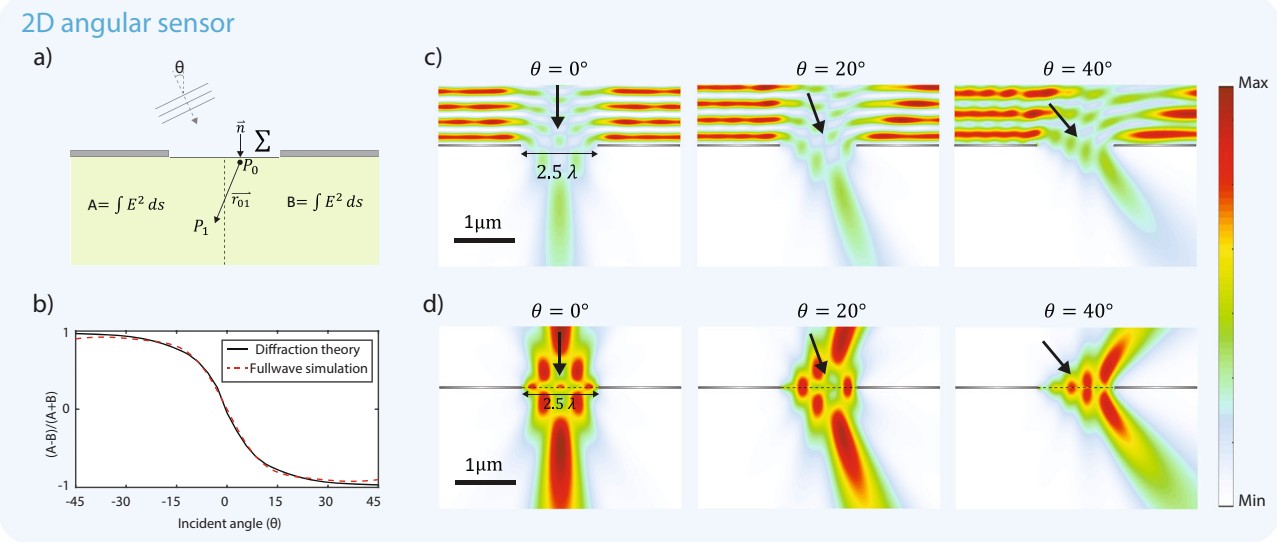

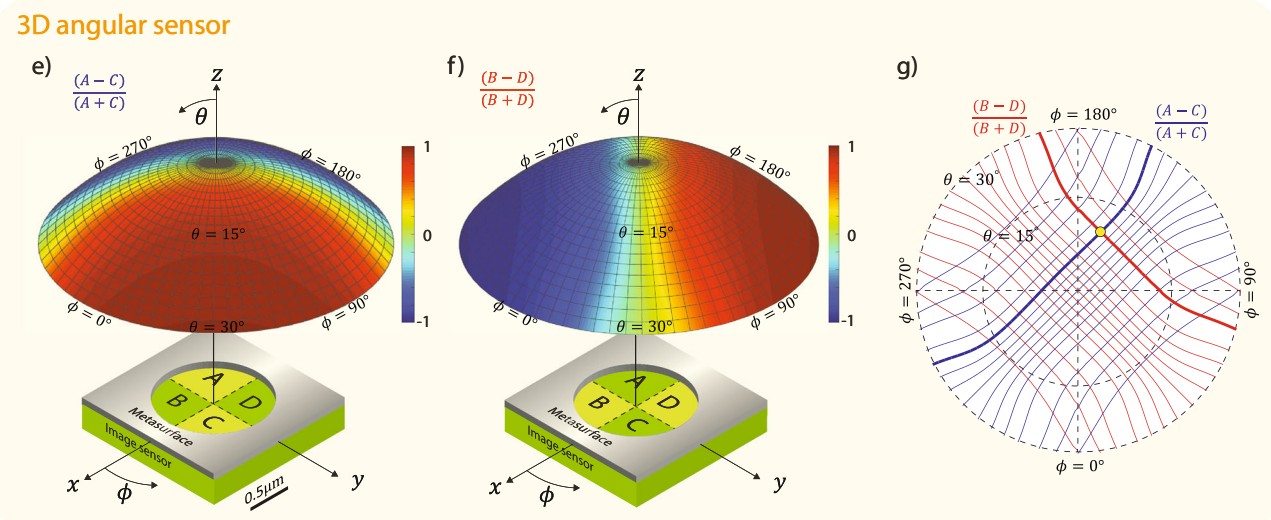

**Fig. 2 Schematic of angle sensing in 2D and 3D space. a** Schematic of single slit aperture for calculating field energy using Rayleigh–Sommerfeld diffraction theory. **b** Plot of total energy ratio between two equally divided regions (A and B) for different incident angles. Calculation based on full-wave simulation and Rayleigh–Sommerfeld diffraction theory are in good agreement. **c** The electric field intensity profile of a plane wave incident on a PEC slab with an open aperture of $2.5\lambda$ width. The intensity profile is a strong function of the incident angle $\theta$. **d** The scattered field by the open aperture. It is calculated by subtracting the incident field from the total field. The incident field is defined as the field without the open aperture but with a perfect PEC reflector. **e** Schematic of angle sensor with an aperture on four photodetectors labeled by A, B, C, and D. The map of energy ratio $\frac{(A-C)}{(A+C)}$ for different incident angles is shown on the hemisphere. Each point on the hemisphere represents one combination of polar angle $\theta$ and azimuthal angle $\phi$. **f** The map of energy ratio $\frac{(B-D)}{(B+D)}$. **g** Contour lines extracted from hemisphere in **e** and **f**. A unique incident angle can be determined when two ratios $\frac{(A-C)}{(A+C)}$ and $\frac{(B-D)}{(B+D)}$ are measured by the four photodetectors.

no field after it. If we create an opening in the perfect metal, there would be field scattered by this opening and the field will be superimposed on top of the background field. The non-uniform field distribution as seen in Fig. 2d is created by such scattering field. This scattered field is directional, depending on the incident angle. It creates angular dependence of the system. Angular response of the actual three-dimensional (3D) structure is calculated using hardware-accelerated full-wave solver, Tidy3D (https://simulation.cloud; see Supplementary Section 2).

Angles in 3D space can be measured similarly. We divide the volume below an aperture into four equal parts as shown in Fig. 2e, each of which is measured by a photodetector. For

any incident wave, we measure two energy ratios defined as $\frac{A-C}{A+C}$ and $\frac{B-D}{B+D}$ where A, B, C, and D represent the energy in each region. Figure 2e, f show the energy ratios for different incident angles. We plot the contours of the two ratios in a polar diagram in Fig. 2g. The combination of two ratios uniquely determines one incident angle as shown by the only crossing point in Fig. 2g.

**Fabrication and characterization of the device.** Next, we discuss the fabrication of the device. Here we use a monochrome complementary metal-oxide-semiconductor (CMOS) image sensor Aptina MT9M001C12STM with a pixel size of 5.2 μm. First, we

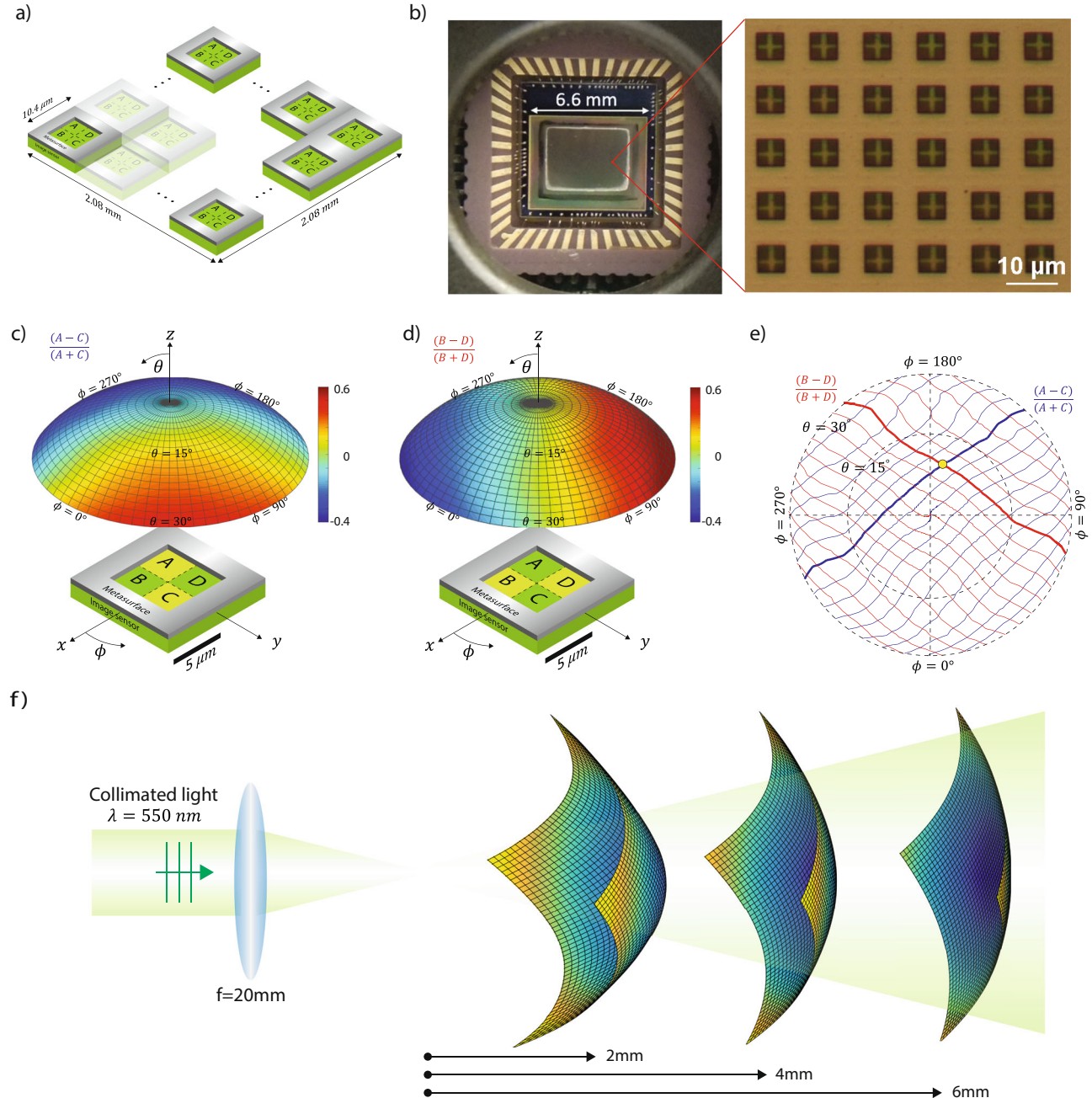

**Fig. 3 Wavefront sensor fabrication and characterization. a** Schematic of a wavefront sensor that consists arrays of unit cell. Each unit cell consists of 2 × 2 pixels. **b** Image of a fabricated wavefront sensor and a microscope image showing its local region where 2 × 2 pixels are partially exposed under each aperture. **c** Measured pixel intensity ratio $\frac{(A-C)}{(A+C)}$ of one unit cell as a function of incident angle shown on a hemisphere. Each point on the hemisphere represents one combination of polar angle $\theta$ and azimuthal angle $\phi$. **d** Measured pixel intensity ratio $\frac{(B-D)}{(B+D)}$ of one unit cell as a function of incident angle shown on a hemisphere. **e** Contour lines extracted from the hemisphere in **c** and **d**. A unique incident angle can be determined when two ratios $\frac{(A-C)}{(A+C)}$ and $\frac{(B-D)}{(B+D)}$ are measured by the four photodetectors. **f** Measurement of a diverging wavefront at three different locations that are 2, 4, and 6 mm away from the focal point. Each grid point represents a measured data. It can be seen that the wavefront is characterized at a high spatial resolution.

removed the micro lens array layer to expose the silicon photo-sensitive materials, which was covered by a thin passivation dielectric. Next, the CMOS Si chip was removed from the chip holder for microfabrication. We used photolithography to define arrays of square patterns on a negative tone photoresist layer. Each square is centered at the touching corner of four neighboring pixels as in Fig. 3a. Its size is 5.2 µm with a periodicity of 10.4 µm. Next, 150 nm-thick aluminum was evaporated directly

onto the image sensor followed by a lift-off process, leaving an aluminum film with square apertures on top of the image sensor. Figure 3b shows a microscope image of a fabricated CMOS Si chip. It is then mounted back to the original chip holder through wire bounding. Each square aperture together with its four pixels underneath constitutes one angle sensor.

The quantum efficiency is reduced by four times compared to the original CMOS image sensor due to light being blocked by

metals. We can significantly improve the quantum efficiency by improving the throughput of light. For example, one design is to use phase aperture instead of the binary aperture, which in principle can improve the light through over 90%.

Large arrays of angle sensors form a wavefront sensor with a very high spatial resolution. In contrast, a Shack–Hartman wavefront sensor uses several hundreds of pixels in each angle sensor and thus has a much lower spatial resolution. See Supplementary Section 10 where we compare conventional Hartmann sensors and our wavefront sensors. They operate at different length scales and thus have different performance.

The sensor needs to be calibrated because of the randomness in the fabrication process and the response of photodetectors. The calibration is done by using a collimated light-emitting diode (LED) light source. Two rotation stages and one linear stage were connected to move the sensor in polar $\theta$ and azimuthal $\phi$ directions with respect to the collimated beam. [Details available in Supplementary Section 3]. The measured pixel intensity ratio $\frac{A-C}{A+C}$ and $\frac{B-D}{B+D}$ of one unit cell obtained from the calibration are shown in Fig. 3c, d on the hemisphere. The incident angle can be uniquely determined as shown by the only crossing point in Fig. 3e.

After calibration, we first characterize a simple wavefront formed by a collimated beam passing through a biconvex lens as in Fig. 3f. We measured the wavefront at three different distances away from the focal point. The curvature of the measured wavefront agrees very well with analytical calculation based on a spherical wavefront originating from the focal point [see details in Supplementary Section 4].

The most significant advantage of this wavefront sensor is its high resolution. The density of spatial sampling is two orders of magnitude higher than most traditional angle-based sensors. For example, here the sampling density is 9246 point/mm$^2$, whereas it is usually 44 points/mm$^2$ for a Shack–Hartmann sensor (e.g., Thorlabs WFS20-5C). A high sampling density makes it possible to resolve fine features in a wavefront. The capability of resolving fine features can be quantified by the maximum phase gradient that can be measured by a wavefront sensor. It is determined by the spatial sampling density and angular dynamic range. Commercial Shack–Hartmann sensors such as Thorlabs WFS20-5C typically have a spatial resolution of 100 μm and an angular dynamic range of < 1°. As a result, its maximum phase gradient at the entrance of the sensor is limited to 0.1 mrad/μm. The sensor shown here has a spatial resolution of 10 μm and an angular dynamic range of 30°. The maximum phase gradient can reach up to 50 mrad/μm, which is 500 times larger than Shack–Hartmann sensor. Such enhanced capability allows angle-based sensors to perform quantitative phase measurement, which is traditionally dominated by the interference-based method.

**Microscopic surface topography**. Next, we use this wavefront sensor to measure microscopic surface topography. Surface topography is widely used in material research and manufacturing, as well as in life science research. Most existing methods are based on interference. Here we show that angle sensors can reach the same level of accuracy and resolution, meanwhile retaining their intrinsic advantages of fast speed and robustness.

The sample to be characterized was a droplet of polymethyl methacrylate (PMMA) polymer on a microscope slide. It was heated under a hot plate to the temperature of 180 °C. As the temperature rose, the polymer changed its state from liquid to solid, creating a wrinkled surface. The goal is to measure its height profile. We choose this sample, because its surface has

abundant microscopic features. The sharply curved morphology with diverse scales can be used to test the performance limit of various surface profilers. For this purpose, we choose a challenging region on the sample with sharp slopes.

One standard metrology tool for this task would be WLI[30]. It scans along the direction perpendicular to sample plane to measure the height. Here we use Zygo NewView 9000 as shown in Fig. 4a. The results are shown in Fig. 4c. The surface varies by 20 μm in height over the measured region. It took 40 s to acquire the image in Fig. 4c.

Next, we measured the same region using our wavefront sensor setup as show in Fig. 4b. It uses a collimated illumination from an LED in a simple 4$f$ system. Two biconvex lenses have a 60 mm focal length. The wavefront sensor directly measures the distribution of angles on the sensor plane, which is then converted to wavefront using zonal estimation described in[7] (see Supplementary Section 5). The surface profile measured by our sensor agrees very well with that from WLI as shown in Fig. 4d. We also show the angular information in Fig. 4e. The direction of arrows represents azimuthal direction of light propagation. The length of arrows represents the magnitude of the polar angle, defined by the angle between the vertical $z$-axis and the direction of light propagation. Polar angles, i.e., the length of arrow, reflect how fast surface heights vary. We use red arrows to indicate a steep slope where surface height varies quickly. These regions can be challenging for WLI to measure. For example, we show another microscopic region (Fig. 4f) that has steeper slopes. The WLI reports white area for these challenging regions (see Supplementary Section 8 for more discussion). On the other hand, our angular sensor, with a high dynamic range of measurable angles, can easily resolve these steep slopes. Figure 4g shows the results with 10 × magnification obtained by replacing the first lens in the 4$f$ system with a 10 × objective lens and second lens with the associated tube lens. The topography obtained from WLI (Fig. 4f) agrees well with that from our sensor in slowly varying regions. The large angles can be observed in Fig. 4h, which is consistent with white regions in Fig. 4f where WLI fails.

A differential interference contrast microscope is known to provide a qualitative image of a surface roughness. A comparison between qualitative image and a quantitative image taken by our wavefront sensor is discussed in Supplementary Section 9.

WLI can realize diffraction-limited resolution in the horizontal plane and can achieve a sub-nanometer resolution in the vertical direction. Although our sensors can also realize diffraction-limited resolution, it is challenging to realize sub-nanometer resolution in the vertical direction. The important advantage of angle-based sensor over WLI is the fast speed. A single topography took 30 ms to acquire the results shown in Fig. 4d. This time is only limited by the frame acquisition time of cameras. Thus, it can be easily improved by using high-speed camera to reach above 1000 Hz. In contrast, the image in Fig. 4c took WLI 40 s due to vertical scanning and stitching.

The fast speed allows us to demonstrate a useful capability: capturing real-time topography at a video-frame rate. This capability can be used for observing temporal evolution of living biological samples or for studying dynamics of materials going through fast changes.

Here we measure the temporal dynamics of a surface during a fast coagulating process. A droplet of PMMA polymer on a transparent microscope slide is heated by a heat gun. Under the blow of air flow and high temperature, its surface goes through rapid change. The surface topography is recorded at 30 frames per second. Figure 5e shows the setup. Figure 5a–d show the

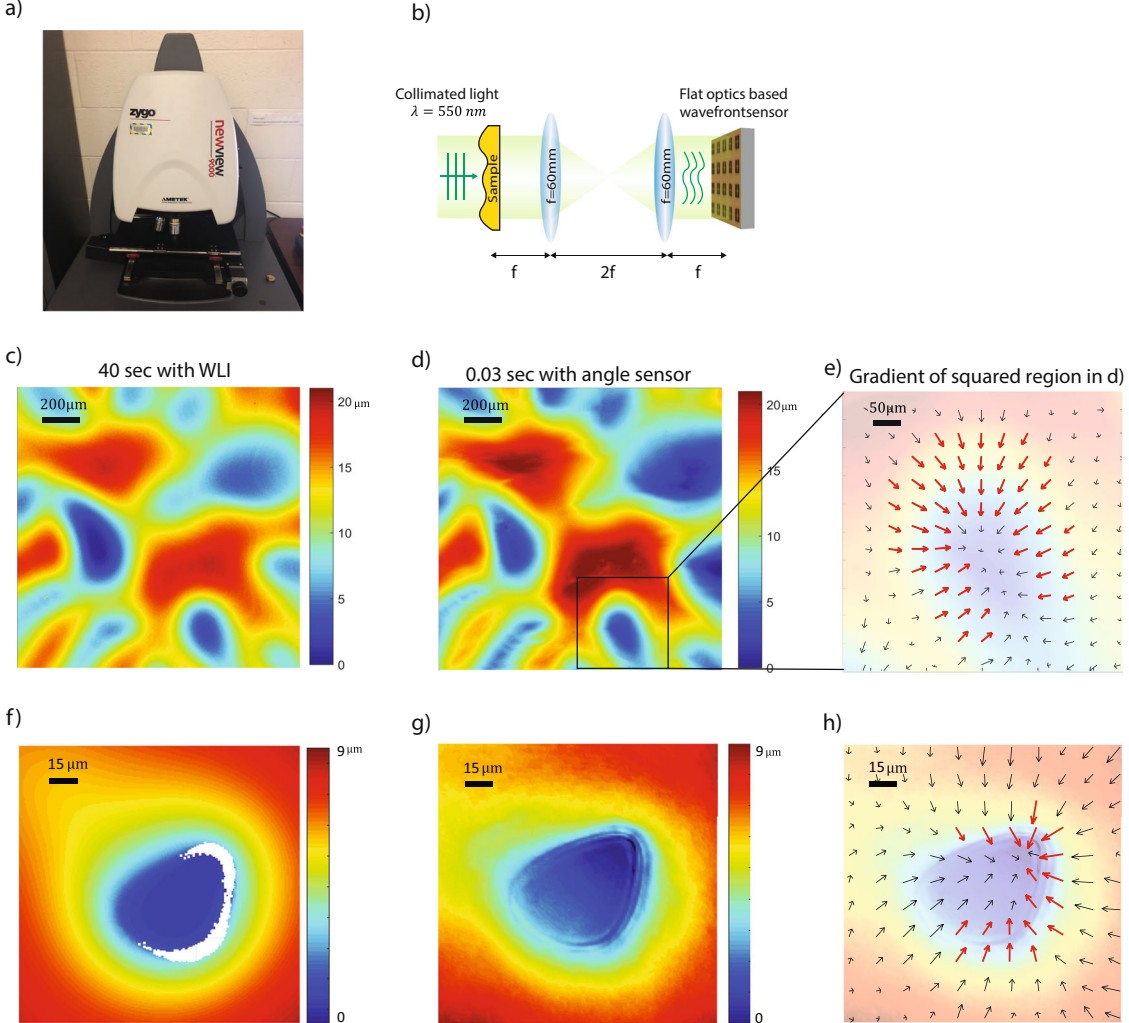

**Fig. 4 Surface profile measurement comparison with white light interferometer. a** Photo of a commercial white light interferometer (Zygo NewView 9000) for surface profile measurement. **b** Optical setup for surface profile measurement using the proposed wavefront sensor where f is the focal length of the lens. **c** Surface profile of a PMMA polymer on a microscope slide measured with a WLI. **d** Same sample as in **c** measured with the proposed wavefront sensor. **e** Close-up view of the cropped region in **d** where the slope of the surface is represented with arrows indicating projected vector components. **f** PMMA polymer surface measured with a WLI equipped with a 10× objective lens. **g** Same sample as in **f** measured with our wavefront sensor equipped a 10× objective lens. **h** Red arrows indicate the region with steep slope where WLI fails to report the height.

time lapse of the surface profile during the phase transition. The video is available in the Supplementary Material. Before the heat is applied, PMMA remains calm and the surface is mostly flat. As the temperature of PMMA increases, the surface will start to show wrinkles, as it begins to coagulate starting from the side where the heat is applied. The surface morphology goes through rapid change as shown in Fig. 5a–d. For future development, it is possible to observe microscopic surface morphology with micro-second temporal resolution by using high-speed camera (Memrecam-ACS-1-M60-40-Data-Sheet.pdf)[31].

The wavefront sensor can also operate under reflection mode by adding a beam splitter to the existing setup. Reflection mode is particularly useful for measuring a sample that is highly reflective. Reflection mode setup and measurement results are discussed in Supplementary Section 6. The sensitivity of the wavefront measurement relies on accurate measurement of incident angle, which is a strong function of the signal-to-noise ratio of the photodetector. We provide brief analysis in Supplementary Section 7.

## Discussion

In conclusion, robust and fast wavefront measurement is of great potential for a number of areas including material characterization, manufacturing, and life science research. Angle-based wavefront sensors have been successful in characterizing slowly varying wavefronts. Here we greatly improve its sampling density and angular dynamic range, to allow it to be used for demanding applications such as quantitative phase imaging. It can complement the interference-based methods that usually dominate this application area. The sensors only involve one deposition and etching step, and thus can be fully integrated into the existing CMOS sensor fabrication process. It could be mass produced at a low cost[32]. The video-frame topography could be useful for material and life scientists to study the temporal dynamics of microscopic samples.

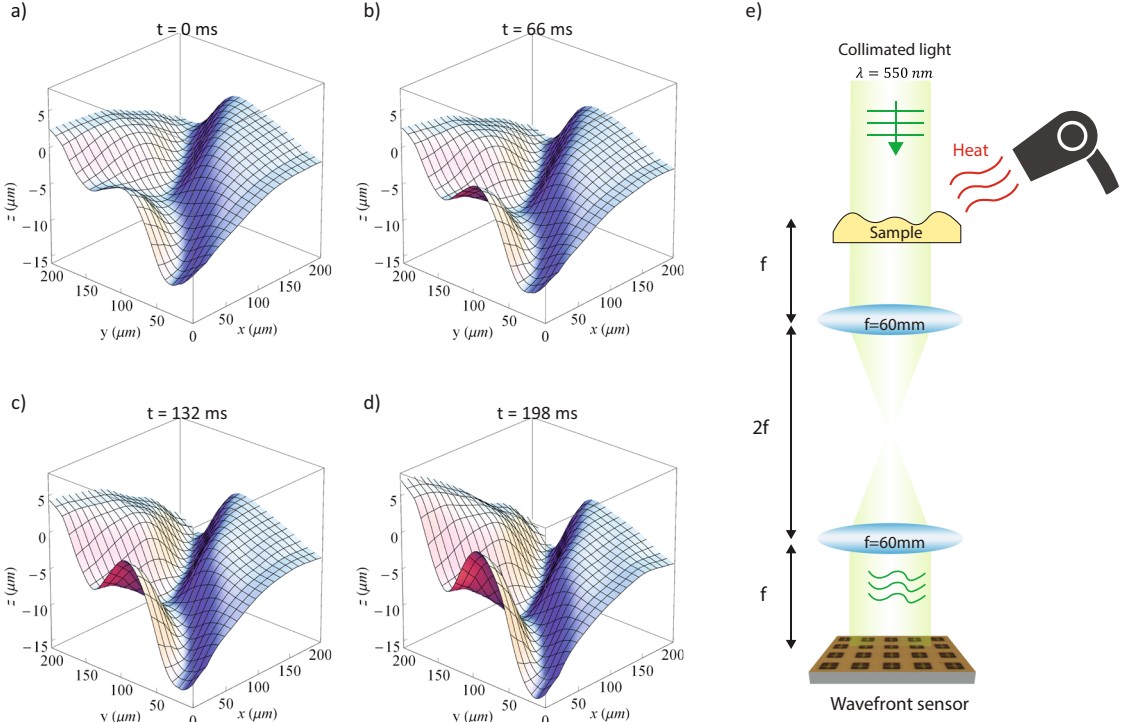

**Fig. 5 Surface topography in video-frame rate. a–d** Selected time lapse of a surface profile captured at a video-frame rate. Sample is a PMMA polymer on a transparent microscope slide that goes through a fast coagulation process, while its temperature increases. **e** Measurement setup for a video-frame rate measurement. Optical setup is same as that in Fig. 4b. A heat gun is used to increase the temperature of a PMMA polymer on a microscope slide.

## Data availability

The data that support the finding of this study are available from the corresponding author upon reasonable request.

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

## Acknowledgements
The work was supported by NSF Career Award (1749050). We acknowledge helpful discussion with Professor Na Ji at UC Berkeley.

## Author contributions
S.Y., L.Y., and Z.Y. conceived the project. S.Y. fabricated the device. S.Y. and J.X. designed the experimental setup and performed the measurements. S.Y., M.Z., and Z.W. performed the simulations. All authors discussed the results and contributed to the writing of the manuscript.

## Competing interests
The authors declare no competing interests.
