## [Peer Review File · Nature Communications]

Reviewer #2 (Remarks to the Author):

The manuscript titled **Angle-based wavefront sensing enabled by the near fields of flat optics** presents a system that allows to obtain high spatial resolution information about a complex field in the visible range. To do so, the authors built a compact system that uses two main elements: a pixelated detector (a CMOS sensor) and a transmission mask consisting of an array of small square apertures. Looking at the energy distribution after the aperture allows to track the incidence angle of the wavefront at each aperture position. Given that the incidence angle is related to the derivative of the wavefront phase, it is possible to use common established techniques in wavefront sensing (numerical integration, modal reconstruction, etc.) to recover the phase distribution of the field.

The manuscript is clear and precise. There is a rigorous description of the fundamentals of the technique based on wave propagation, where the relationship between the wavefront incidence angle and the energy distribution after the aperture is derived. These foundations are tested in real experiments and show a very good agreement. Experimental results are shown, and serve as a good benchmark of the capabilities of the technique with regard to spatial resolution, dynamic range, and sensitivity. Figures are clear, and there is enough information in the manuscript and the supplement in order for other researchers to replicate and further develop the technique presented in the text.

However, in its current state, I cannot recommend the manuscript for publication in Nature Communications. I list my concerns here:

The main benefit that the authors list in the manuscript is that the system presented here outperforms the capabilities of conventional Shack-Hartmann wavefront sensors (SHWS) sensors. The spatial resolution in SHWS is limited by the physical size of the microlenses, which can hardly be fabricated with sizes below a hundred microns. On the other hand, the apertures used in this sensor can be built by using photolithography, which allows much smaller sizes. In the end, this increases a lot the spatial sampling, providing much better spatial resolution. Also, SHWS have hard tradeoffs between spatial resolution, sensitivity, and dynamic range. Again, the sensor presented here goes far beyond common limits for SHWS technology. However, there are other aspects that were not discussed in the paper. For example, how does this sensor compare regarding photon efficiency? Microlens arrays can be built with very high filling factors (almost 100%), and the mask shown by the authors seems to have a filling factor of about 50%. Furthermore, this system uses just a small area of each pixel present in the detector to obtain information. I wonder how this affects the quantum efficiency, and which illumination level regimes could be explored. In order to make a more robust comparison, a deeper study should be realized.

Moreover, while the comparison with SHWS makes sense (after all, they are the standard in many fields), the system presented here is extremely close to a Hartmann wavefront sensor (which uses pinhole arrays instead of microlenses). In fact, while reading the manuscript it was quite hard for me to grasp where was the novelty between the system shown here and Hartmann sensors that have been already used in several spectral ranges (mainly in x-ray and UV regions [1,2], but also in astronomy applications). After all, the energy distribution calculation procedure seems to be more or less the same as centroid calculation in conventional Hartmann sensors: in the end, just 4 pixels need to be used to calculate the centroid of an intensity distribution with very good precision, as is usually done in quadrant lateral position detectors. In order to

assess the novelty and impact of this manuscript, a thorough discussion about the similarities of this technique and Hartmann sensors should be done, stating where the novelty resides. Otherwise, the ideas shown here might seem just a technical improvement over already established approaches.

There is also a minor detail regarding Fig.1.c:

Caption: c) The near-field distribution behind the aperture. The energy distribution is a strong function of the incident angle.

In this part of the figure, the authors show the energy distribution before and after the aperture for three different incidence angles. There are some aspects that I did not fully understand, and I could not find an explanation on the text or the caption. I understand that the wave is propagating from top to bottom, but if the illumination consists of a plane wave, I do not get why we see a non-uniform energy distribution before the aperture. Is that due to reflection effects by the aperture? If that is the case, I also do not understand the effects on the energy when tilting the illumination. I think a little bit more information on this should be provided.

[1] Daniele Cocco, Mourad Idir, Daniel Morton, Lorenzo Raimondi, Marco Zangrando, Advances in X-ray optics: From metrology characterization to wavefront sensing-based optimization of active optics, Nuclear Instruments and Methods in Physics Research Section A: Accelerators, Spectrometers, Detectors and Associated Equipment, Volume 907,2018

[2] Pascal Mercère, Philippe Zeitoun, Mourad Idir, Sébastien Le Pape, Denis Douillet, Xavier Levecq, Guillaume Dovillaire, Samuel Bucourt, Kenneth A. Goldberg, Patrick P. Naulleau, and Senajith Rekawa, "Hartmann wave-front measurement at 13.4 nm with $\lambda\text{EUV}/120$ accuracy," Opt. Lett. 28, 1534-1536 (2003)

Reviewer #3 (Remarks to the Author):

The authors present a quite interesting method of using the diffraction from the aperture to measure the angle of incident fields. Most significantly, this device only requires four pixels to capture the angular information, having better resolution than the traditional Shack-Hartmann sensor. I think the idea is valid and can have a good impact.

However, I have two main problems with this work:

- The schematic diagram in figure 1 demonstrates the angular dependence of the small metal aperture (1.375 μm in diameter). Besides, the authors show more angular dependence in the SI, where the largest diameter is 2 μm . When the aperture gets larger, the angular dependence gets weaker. However, in figure 2, the authors are showing devices with a 5.2 μm aperture diameter, which is much bigger than 2 μm (and I won't call this metasurface at this scale). When the aperture size is almost 10 times the wavelength, intuitively I don't expect much angular dependence. I think more simulation results here will be helpful. What is the distance between the metal aperture and the actual pixel? Is it possible to show a cross-section view of the full-wave simulation (FEM, FDTD, ...) of the actual device with angular dependency?
- In figure 3, the authors compare the surface profiling performance of commercial WLI and their newly proposed method. In general, WLI is a very precise metrology tool, which can achieve wavelength level spatial resolution and measure tens of micrometer in height. Here in the experiment, the quality of the WLI measurement seems not good enough, not showing its true resolution capability. I think it is probably due to the sample selection and low NA. Can you repeat this experiment with a nano-fabricated sample with an SEM image as the ground truth?

I therefore cannot recommend accepting the paper for publication in its present form. A major revision needs to be done to address the aforementioned issues.

- Since the authors claim they can achieve the same level of accuracy and resolution as commercial WLI, can you also comment on the (X, Y, Z) resolution of this method? What is the resolution limit of this method, not limited to a lab-fabricated device? Given the resolution of this method, what specific application can it be used for?

RESPONSE TO REVIEWER COMMENTS

Reviewer #2 (Remarks to the Author):

Reviewer: The manuscript titled Angle-based wavefront sensing enabled by the near fields of flat optics presents a system that allows to obtain high spatial resolution information about a complex field in the visible range. To do so, the authors built a compact system that uses two main elements: a pixelated detector (a CMOS sensor) and a transmission mask consisting of an array of small square apertures. Looking at the energy distribution after the aperture allows to track the incidence angle of the wavefront at each aperture position. Given that the incidence angle is related to the derivative of the wavefront phase, it is possible to use common established techniques in wavefront sensing (numerical integration, modal reconstruction, etc.) to recover the phase distribution of the field.

The manuscript is clear and precise. There is a rigorous description of the fundamentals of the technique based on wave propagation, where the relationship between the wavefront incidence angle and the energy distribution after the aperture is derived. These foundations are tested in real experiments and show a very good agreement. Experimental results are shown, and serve as a good benchmark of the capabilities of the technique with regard to spatial resolution, dynamic range, and sensitivity. Figures are clear, and there is enough information in the manuscript and the supplement in order for other researchers to replicate and further develop the technique presented in the text.

However, in its current state, I cannot recommend the manuscript for publication in Nature Communications. I list my concerns here:

The main benefit that the authors list in the manuscript is that the system presented here outperforms the capabilities of conventional Shack-Hartmann wavefront sensors (SHWS) sensors. The spatial resolution in SHWS is limited by the physical size of the microlenses, which can hardly be fabricated with sizes below a hundred microns. On the other hand, the apertures used in this sensor can be built by using photolithography, which allows much smaller sizes. In the end, this increases a lot the spatial sampling, providing much better spatial resolution. Also, SHWS have hard tradeoffs between spatial resolution, sensitivity, and dynamic range. Again, the sensor presented here goes far beyond common limits for SHWS technology.

Response: We thank the reviewer for his/her support and encouragement.

Reviewer: However, there are other aspects that were not discussed in the paper. For example, how does this sensor compare regarding photon efficiency? Microlenses arrays can be built with very high filling factors (almost 100%), and the mask shown by the authors seems to have a filling factor of about 50%. Furthermore, this system uses just a small area of each pixel present in the detector to obtain information. I wonder how this effects the quantum efficiency, and which illumination level regimes could be explored. In order to make a more robust comparison, a deeper study should be realized.

Response: Thanks for the comments. The quantum efficiency is reduced by four times compared to the original CMOS image sensor which we used due to light being blocked by metals. There is

no fundamental limit that prevents this concept from realizing a high efficiency through improving the light throughput. This is an area that we are working to improve. In the next generation design, we are able to greatly improve the light throughput by replacing binary mask with phase mask. The design is illustrated below.

Instead of light-blocking binary mask (metallic aperture as shown in Fig. R1a Left), here we use phase mask based on dielectric materials (Fig. R1a Right). Light passing through the dielectric material and air undergo longer optical path, resulting in additional phase. When the phase delay is π , the destructive interference between light passing through dielectric mask and air will redistribute the energy of transmitted light, creating hotspots in the near field of energy distribution. Because of the interference, the positions of these hotspots are very sensitive to incident angle, enabling angle-sensing capability. More importantly, the efficiency of such mask can be higher than 90%. As a specific example, we consider a dielectric mask with a dielectric constant of 2.25. Figure R1b shows the energy distribution of the dielectric phase mask under different incident angle. The energy distributions clearly show angular dependence, and the transmission of such mask reaches almost 97%.

Fig. R1. a) Schematics of a binary mask and a phase mask. The binary mask blocks light transmission. In contrast, the phase mask allows light passing through it, but with a phase delay of π . b) Example of angle dependent intensity distribution for three different incident angles 0° , 5° and 10° . The width and thickness of the mask are 2.5λ and λ , respectively.

We have revised the manuscript to comment on the quantum efficiency and add some discussion to the supplementary

Reviewer:

Moreover, while the comparison with SWHS makes sense (after all, they are the standard in many fields), the system presented here is extremely close to a Hartmann wavefront sensor (which uses pinhole arrays instead of microlenses). In fact, while reading the manuscript it was quite hard for me to grasp where was the novelty between the system shown here and Hartmann

sensors that have been already used in several spectral ranges (mainly in x-ray and UV regions [1,2], but also in astronomy applications). After all, the energy distribution calculation procedure seems to be more or less the same as centroid calculation in conventional Hartmann sensors: in the end, just 4 pixels need to be used to calculate the centroid of an intensity distribution with very good precision, as is usually done in quadrant lateral position detectors. In order to assess the novelty and impact of this manuscript, a thorough discussion about the similarities of this technique and Hartmann sensors should be done, stating where the novelty resides. Otherwise, the ideas shown here might seem just a technical improvement over already established approach.

Response:

Fig. R2. a) Schematic of our wavefront sensor with extremely small mask and short distance between mask and sensor layer. b) Schematic of conventional Hartmann sensor with extremely large aperture and mask to sensor distance. c) Example of raw measurement data captured with our wavefront sensor. d). Raw measurement data of conventional Hartmann sensor cited from [1] only showing 15x15 spatial resolution.

Previously, people used apertures to measure wavefronts in EUV [2] and X-ray [1]. This is due to the lack of lens in these wavelength ranges. However, being micro-lens in the visible range or apertures in the EUV/X-ray range, traditional Hartmann sensors follow the same operating principle, and it is different with our sensor in one important aspect: **length scale**

The difference in length scale leads to different regimes for the wave physics and the resulting performance metric is also drastically different. The aperture in EUV/X-ray, are 1000 – 10,000 times of wavelength. The distance between the aperture and the sensor plane is even larger (Fig. R2b). In contrast, all length scale in our system is all around wavelength scale (Fig. R2a). This difference dictates that the two systems explore different physics: the former

primarily relies on **ray optics with far-field diffraction correction**. The latter, i.e. our sensor, needs **full wave electrodynamics and exploit near-field energy distribution**. Because we explore a quite different length scale and primary physical mechanism, the new system can realize a performance that is **orders of magnitude better in both spatial resolution and angular dynamic range**. Consequently, this performance improvement enables the angle-based approach to be used for quantitatively phase imaging, a new area of angle-based sensor. This further enables a new capability for real-time video recording of microscopic phase front. Such new capability is highly desired for bio imaging and material characterization.

There is also significant difference in the measured data which can be seen in Fig. R2(c) and (d). We have added new discussion in both the main manuscript and the supplementary to further improve this point.

Reviewer: There is also a minor detail regarding Fig.1.c:

In this part of the figure, the authors show the energy distribution before and after the aperture for three different incidence angles. There are some aspects that I did not fully understand, and I could not find an explanation on the text or the caption. I understand that the wave is propagating from top to bottom, but if the illumination consists of a plane wave, I do not get why we see a non-uniform energy distribution before the aperture. Is that due to reflection effects by the aperture? If that is the case, I also do not understand the effects on the energy when tilting the illumination. I think a little bit more information on this should be provided.

Response: Thank you for pointing this out.

We can understand the field by using the total-field scattered-field method. Without the aperture, a perfect metal mask will simply reflect the incident wave (Fig. R3a). We have interference built up in front of the perfect metal and no field after it. When we create an opening in the perfect metal, there would be field scattered by this opening (Fig. R3b) and the field will be superimposed on top of the background field (Fig. R3c).

The scattered field of the aperture is now computed using the technique of total-field and scattered-field. They are plotted below in Figure R3b. The non-uniform energy distribution as seen in the figure in the manuscript is created by such scattering field. This scattered field is very directional, depending on the incident angle. It is this scattered field that creates angular dependence of the system. We have now included this discussion in the main manuscript.

Fig. R3. a) Electric field intensity profile of a planewave incident on a PEC slab. b) Scattered field created by a planewave incident on a PEC slab with an aperture of 2.5λ width. c) Total-field profile which is summation of background field in a) and scattered field in b). Black arrow indicates the incident angle of light.

We have included the new figure in the revised manuscript.

Reviewer #3 (Remarks to the Author):

Reviewer: The authors present a quite interesting method of using the diffraction from the aperture to measure the angle of incident fields. Most significantly, this device only requires four pixels to capture the angular information, having better resolution than the traditional Shack-Hartmann sensor. I think the idea is valid and can have a good impact.

Response: We thank the reviewer for his/her support and encouragement.

Reviewer: • The schematic diagram in figure 1 demonstrates the angular dependence of the small metal aperture ($1.375\ \mu\text{m}$ in diameter). Besides, the authors show more angular dependence in the SI, where the largest diameter is $2\ \mu\text{m}$. When the aperture gets larger, the angular dependence gets weaker. However, in figure 2, the authors are showing devices with a $5.2\ \mu\text{m}$ aperture diameter, which is much bigger than $2\ \mu\text{m}$ (and I won't call this metasurface at this scale). When the aperture size is almost 10 times the wavelength, intuitively I don't expect much angular dependence. I think more simulation results here will be helpful. What is the distance between the metal aperture and the actual pixel? Is it possible to show a cross-section view of the full-wave simulation (FEM, FDTD, ...) of the actual device with angular dependency?

Response:

For figure 2 where a $5.2\ \mu\text{m}$ aperture was used for an actual device, silicon layer (array of pixels) is below the $5.2\ \mu\text{m}$ aperture. Thus the distance between metal aperture and the actual pixel can be considered almost zero.

Here we simulate the cross section of $5.2\ \mu\text{m}$ wide aperture in 2D FEM simulation. The underlying physics are the same. Figure R4a shows the setup of the simulation. Perfect matching layers are used in all directions. Aluminum masks with $5.2\ \mu\text{m}$ wide aperture are placed on top of silicon layers. For clarity, here we assume silicon is lossless. Figure R4b shows the intensity distribution of the scattered field by the aperture under different incident angle. Angular dependence of intensity distribution can be clearly seen, even though the aperture is $5.2\ \mu\text{m}$.

Fig. R4 (a) Schematic of the simulation. (b) Intensity distributions under different incident angle.

Reviewer:

- In figure 3, the authors compare the surface profiling performance of commercial WLI and their newly proposed method. In general, WLI is a very precise metrology tool, which can achieve wavelength level spatial resolution and measure tens of micrometer in height. Here in the experiment, the quality of the WLI measurement seems not good enough, not showing its true resolution capability. I think it is probably due to the sample selection and low NA. Can you repeat this experiment with a nano-fabricated sample with an SEM image as the ground truth?

Response:

In general, when the slope of the surface to be measured is large, WLI can fail, showing a blank region. This is quite common observation in using WLI. The reviewer is correct that a low NA contributes to the effect. Being sensitive to NA is a disadvantage of WLI. However, by using latest WLI and fine tune the imaging configuration can minimize such failed regions.

The data shown previously was done in Zygo NewView 6300 model, which is many years old. When we went back to perform the experiment after Covid shut-down, the latest model Zygo NewView 9000 become available in our public facility. The handling of large slope is highlighted as one of the major upgrades [3]. We were able to perform the same measurement with the latest advanced WLI. Some of the white regions, but not all, can be removed due to the instrument upgrade. The results agree very well with our results. The results are shown below.

Fig. R5. Surface profile measured with Zygo NewView 9000

We have replaced the old results with latest WLI result and focus on the comparison of latest commercial WLI, which can serve as the ground truth. However, when the sample is really sharp in slope as that in Fig. 3f, we still can observe the failed region even in this upgraded WLI system.

Separately, we attempted to perform nanofabrication and SEM of new surface sample. The first author recently graduated and COVID has impacted our training process. We were not able to do SEM in a timely fashion. We hope the newer result WLI can serve a similar purpose.

Reviewer: Since the authors claim they can achieve the same level of accuracy and resolution as commercial WLI, can you also comment on the (X, Y, Z) resolution of this method? What is the resolution limit of this method, not limited to a lab-fabricated device? Given the resolution of this method, what specific application can it be used for?

Response:

The resolution in X, Y dimension is governed by sparrow resolution limit, $\Delta x = \Delta y = 0.47\lambda/NA$. For example, resolution in X, Y dimension of a commercial white light interferometer (Zygo NewView 9000) ranges from 0.43 to 11.6 μm which is objective lens dependent. Based on sparrow resolution limit, lateral resolution of our device is 0.86 μm when coupled with a 10X objective lens with 0.3 NA. When higher magnification is used, e.g., 100X objective with 0.8 NA, lateral resolution can be as low as 0.32 μm .

The resolution in z-direction is provided as 0.1nm in commercial WLI (Zygo NewView 6300). This would require specific sample type and very low noise detector. The device demonstrated here cannot reach 0.1nm resolution due to the noise of the detectors used.

Since our device is a phase front sensor, the z-resolution is calculated through a somewhat complex numerical method [4]. The analytical form of the resolution limit is not directly available. It is generally determined by the detector noise and the accuracy of phase measurement. The phase front accuracy can be calculated as below.

The minimum detectable angle $\delta\theta$ of the wavefront sensor can be expressed as

$$\delta\theta = \Delta R \cdot \frac{d\theta}{dR} \quad (1)$$

where R is the pixel intensity ratio between two neighboring pixels. If we assume R has a linear response up to $\theta = D$ as shown in Fig. S7a, $d\theta/dR$ can be expressed as

$$\frac{d\theta}{dR} = \frac{D}{R_{max} - 1} \quad (2)$$

where D is the maximum angle that wavefront sensor has a linear response and R_{max} is the maximum pixel intensity ratio at D degree.

Here, we show how minimum detectable angle around normal incidence can be calculated. Since $R = P_1/P_2$ where P_1 and P_2 are pixel intensity of two neighboring pixels, ΔR can be expressed as:

$$\Delta R = \left| \frac{dR}{dP_1} \right| \Delta P_1 + \left| \frac{dR}{dP_2} \right| \Delta P_2 = \left| \frac{1}{P_2} \right| \Delta P_1 + \left| \frac{P_1}{P_2^2} \right| \Delta P_2 \quad (3)$$

Since $P_1 \approx P_2$ when light is normal incident, Eq. S5 can be written as

$$\Delta R = \left| \frac{1}{P_1} \right| \Delta P_1 + \left| \frac{1}{P_2} \right| \Delta P_2 = \frac{2}{SNR} \quad (4)$$

Thus, substituting Eq. 2 and 4 into Eq. 1, $\delta\theta$ can be expressed as

$$\delta\theta = \frac{2}{SNR} \cdot \frac{D}{R_{max} - 1} \quad (5)$$

If we assume $SNR = 45$ dB and use $R_{max} = 1.1$ and $D = 5^\circ$ which is based on our experimental results in Fig. R6b, $\delta\theta$ can be calculated as

$$\delta\theta = \frac{2}{10^{4.5}} \cdot \frac{5^\circ}{1.1 - 1} \approx 0.0032^\circ \quad (6)$$

Fig. R6. a) Pixel intensity ratio (R) of two neighboring pixels as a function of incident angle (θ). b) Measured pixel intensity ratio of two neighboring pixels. The neighboring pixels were randomly selected within the fabricated wavefront sensor.

Resolution in Z dimension is related to the minimum detectable angle of our device. As discussed in SI section 7, minimum detectable angle $\delta\theta \approx 0.0032^\circ$. This angle can be converted to 1.2 nm resolution in Z dimension. This theoretical resolution limit is about 10 times worse than the theoretical limit of WLI.

On application side, we do not expect our device can reach the sub-nanometer resolution that is claimed by WLI. But its advantages is in offers ultra-fast measurement thanks to its single shot measurement capability. Such capability of measuring microscopic morphology with real-time can be used to study temporal dynamics that is difficult to measure before. More broadly, the device beyond to the category of quantitatively phase imaging, which has broad application in biomedical application [5]. Furthermore, our new capability of video-frame recording could further be used to for temporal dynamics of cell interaction or monitoring red blood cell.

References

- [1] Daniele Cocco, Mourad Idir, Daniel Morton, Lorenzo Raimondi, Marco Zangrando, Advances in X-ray optics: From metrology characterization to wavefront sensing-based optimization of active optics, Nuclear Instruments and Methods in Physics Research Section A: Accelerators, Spectrometers, Detectors and Associated Equipment, Volume 907,2018
- [2] Pascal Mercère, Philippe Zeitoun, Mourad Idir, Sébastien Le Pape, Denis Douillet, Xavier Levecq, Guillaume Dovillaire, Samuel Bucourt, Kenneth A. Goldberg, Patrick P. Naulleau, and Senajith Rekawa, "Hartmann wave-front measurement at 13.4 nm with λ EUV/120 accuracy," Opt. Lett. 28, 1534-1536 (2003)
- [3] <https://zygo.de/?/en/met/profilers/newview9000/>
- [4] Soongyu Yi, Ming Zhou, Zongfu Yu, Pengyu Fan, Nader Behdad, Dianmin Lin, Ken Xingze Wang, Shanhui Fan, and Mark Brongersma, "Subwavelength angle-sensing photodetectors inspired by directional hearing in small animals", Nature Nanotech 13, 1143–1147 (2018).
- [5] YongKeun Park, Christian ,epeursinge and Gabriel Popescu, "Quantitative phase imaging in biomedicine", Nature Photon 12, 578–589 (2018).

REVIEWERS' COMMENTS

Reviewer #2 (Remarks to the Author):

My two main concerns on the first submission were the lack of a discussion regarding the similarities of the technique showed in the paper and common Hartmann sensors, and more information regarding the limitations of the approach (mainly quantum efficiency and the level of illumination that could be achieved).

On the Hartmann side, the authors added a discussion in the supplement comparing both techniques, so now it is way clearer which are the differences between them, and also what is the benefit of their approach. Regarding illumination levels / quantum efficiency, the authors clarified that their design, indeed, suffers when reducing the active area of each pixel by using the metallic mask. However, they introduced a novel design using phase masks that should not reduce the number of arriving photons to the sensor, thus evading this problem.

Given that both complains were correctly addressed, and some other minor clarifications were added to the text, I recommend the publication of the manuscript.

Reviewer #3 (Remarks to the Author):

The author has addressed my concerns reasonably. I would recommend this paper published on Nature Communication, with a minor suggestion.

I would suggest it is a "must" to include the FEM simulation and energy ratio plot with the actual device dimension, at least in the supplementary (the simulation results shown in the paper all use much smaller apertures). Since the aperture is pretty large, it would lower the system SNR overall. Especially, In the rebuttal article, the scale bar and the dimension of the simulation need to be enlarged to distinguish the energy ratio. 0 degree and 20 degree cases are quite identical. Adding dielectric spacing layer (as real device does) in the simulation may improve the energy ratio.

Other than this minor suggestion, I think the paper is well written.

RESPONSE TO REVIEWER COMMENTS

Reviewer #2 (Remarks to the Author):

Reviewer: My two main concerns on the first submission were the lack of a discussion regarding the similarities of the technique showed in the paper and common Hartmann sensors, and more information regarding the limitations of the approach (mainly quantum efficiency and the level of illumination that could be achieved).

On the Hartmann side, the authors added a discussion in the supplement comparing both techniques, so now it is way clearer which are the differences between them, and also what is the benefit of their approach.

Regarding illumination levels / quantum efficiency, the authors clarified that their design, indeed, suffers when reducing the active area of each pixel by using the metallic mask. However, they introduced a novel design using phase masks that should not reduce the number of arriving photons to the sensor, thus evading this problem.

Given that both complains were correctly addressed, and some other minor clarifications were added to the text, I recommend the publication of the manuscript.

Response: We thank the reviewer for his/her support and encouragement.

Reviewer #3 (Remarks to the Author):

Reviewer: The author has addressed my concerns reasonably. I would recommend this paper published on Nature Communication, with a minor suggestion. I would suggest it is a "must" to include the FEM simulation and energy ratio plot with the actual device dimension, at least in the supplementary (the simulation results shown in the paper all use much smaller apertures). Since the aperture is pretty large, it would lower the system SNR overall. Especially, In the rebuttal article, the scale bar and the dimension of the simulation need to be enlarged to distinguish the energy ratio. 0 degree and 20 degree cases are quite identical. Adding dielectric spacing layer (as real device does) in the simulation may improve the energy ratio. Other than this minor suggestion, I think the paper is well written.

Response: We thank the reviewer for his/her suggestion. 3D simulation result of the actual device dimension including a dielectric spacing layer is added to Supplementary Note 2.